# Profile and outcome of multiple myeloma with and without HIV treated at a tertiary hospital in KwaZulu-Natal, South Africa

**Lungisile Hildegard Chili**[1,2]*, **Irene Mackraj**[1], **Nadine Rapiti**[1,2]

**1** Haematology Department, School of Laboratory Medicine and Medical Sciences, College of Health Sciences, University of KwaZulu Natal, Durban, KwaZulu-Natal, South Africa, **2** Haematology Department, National Health Laboratory Services, Durban, KwaZulu-Natal, South Africa

* drlmatha@gmail.com

**Data Availability Statement:** All relevant data are within the paper and its Supporting Information files.

## Abstract

### Objectives

To profile the outcome of multiple myeloma (MM) patients treated at a South African tertiary hospital in KwaZulu-Natal (KZN) and to compare MM in HIV-negative patients and MM in people living with HIV (PLWH).

### Methods

A retrospective analysis of patients with MM was conducted over 5 years (2015–2020). Patient demographics, presenting complaints, symptom duration, disease stage, molecular profile, treatment, and survival data were captured. Statistical analysis was conducted using R Statistical software of the R Core Team, 2020, version 3.6.3.

### Results

135 patients; 79% (n = 106) HIV-negative and 21% (n = 29) PLWH were investigated. 54% (n = 74) females and 57% (n = 76) 51–70-year-olds. The 40-50-year-old patient group had a significantly higher proportion of PLWH (p = 0.032). Pathological fractures were the commonest presenting complaint, 47% (n = 57 and 49% (n = 49) had International Staging System, stage III disease. Fluorescent in-situ hybridization (FISH) MM profiling was completed in 58% (n = 78). Positivity for del 11q22 was found in 23.7% (n = 14) with significantly more HIV-negative patients having the mutation (p = 0.027). Overall, 42.2% (n = 57) achieved 2-year overall survival (OS). There were no significant differences in treatment (p = 0.926) and 2-year survival outcome (p = 0.792) between the two groups.

### Conclusion

The incidence of HIV in newly diagnosed MM patients in KZN was increasing. KZN patient profile differed from other reports by showing female predominance but was similar in advanced-stage presentation and bone fracture predominance. Statistically significant

**Funding:** The authors acknowledge funding from the Bristol Myers Squibb Foundation Global Cancer Disparities Africa Collaboration for Multiple Myeloma ABCDES (Awareness and Building Capacity to Diagnose Early and Support) Program (grant number GRANT95856 95555) awarded to LHC.

**Competing interests:** The authors have declared that no competing interests exist.

differences between the HIV-negative patients and PLWH were observed in age distribution and mutational landscape. Further studies are required in this area.

## Introduction

Multiple myeloma (MM) is a bone marrow-based plasma cell malignancy, with the clonal plasma cells producing a monoclonal protein, usually detected in the blood and or urine [1]. It is the second commonest haematological malignancy [2, 3]. The incidence in Africans is 2-fold higher than in Caucasians [4, 5]. The average age at diagnosis is 65 years, and it has a slight male predominance with a M:F ratio of 1.1:1 [6, 7]. The average age at diagnosis in first-world populations is 69 years [5, 7, 8]. However, African and Middle-Eastern researchers report a mean age ranging from 58 to 62 years at presentation [5–7].

In the year 2020, global cancer statistics reported a total of 176 404 new MM cases and 117 077 deaths from MM worldwide [9]. Further to this, a mortality rate of 1.8/100 000, and an incidence rate of 2.3/100000 were also recorded in 2020 [9]. Globally the incidence of MM is increasing due to longer general life expectancy [10, 11]. Death in MM is also not uncommon however during first-line treatment it is often not solely due to disease progression, other factors play a role [12]. In later treatment lines most deaths are due to progressive disease [12].

A Sub-Saharan African analysis found that anaemia and renal failure at baseline were associated with shorter survival and that the most common presentation of MM is bone pain, with spinal involvement, followed by anaemia [13]. In South Africa, the incidence of MM is increasing and is higher in males than females [14].

As of 2021, South Africa has 7.8 million people living with HIV (PLWH), and 72% of those are on antiretroviral treatment (ART) [15]. Data suggests that T-cell depletion and immune deficiency secondary to HIV disease may have an essential role in the increased development of MM in HIV-positive patients [16]. There is local evidence that PLWH are diagnosed with MM at a younger age, and present with more advanced disease or aggressive types of plasma cell dyscrasias [17, 18]. It has also been reported that some ART protease inhibitors (PIs) are effective in MM treatment, functioning similarly to the Proteosome Inhibitor therapy known to be effective in MM [19]. In KwaZulu-Natal (KZN), a province in South Africa, Atazanavir (ART-PI) is an available second-line treatment for HIV disease [20]. This drug requires motivation, and certain criteria specific to HIV disease must be met for patients to access it. Thus, patients in KZN are unlikely to receive this PI upfront [19, 20]. HIV is, however, a priority program, and interventions geared towards this disease could be harnessed in favour of better outcomes in MM [21]. Thus, there is a need for studies in South Africa to determine the prevalence and impact of HIV and its treatment on MM outcomes.

Despite recent advances, MM remains largely incurable [2]. New therapies pioneered during the past two decades have improved both progression-free survival (PFS) and overall survival (OS) for MM patients [2, 22]. Although there has been an improvement in survival, contemporary outcomes for unselected MM patients have been difficult to determine precisely, and many of these newer therapies are costly and not readily available for patients in the public health sector in South Africa [23].

Currently, prognostic variables in MM are based on the Revised International Staging System (RISS) for MM, and include the albumin level, Beta 2 microglobulin ($\beta_2$M), LDH level and cytogenetic or fluorescent in situ hybridisation (FISH) profiling [24–26]. Owing to physician factors, limited testing repertoire and financial constraints, most patients in developing

countries have limited risk profiling rather than that as suggested by international guidelines [13].

Thus far, national data reports from South Africa on epidemiology, incidence, prevalence, prognostic variables and HIV in multiple myeloma patients refer to single-centre experiences in which most patients are state patients [4, 6, 27]. There is no representation of MM patients from the province of KZN. With the advent of novel therapeutics and diagnostic procedures, it is unclear whether the current treatment strategies are effective or appropriate in KZN. Thus, there is a need for inclusive and updated information and profiling of the population to inform policy, focus resources and improve patient outcomes.

## Method

### Study area and period

The study was conducted at King Edward VIII Hospital (KEH), located in Durban, KZN, South Africa, from July 2015 to June 2020. KEH is a tertiary, academic hospital that is the major first-line treatment-centre for public patients with MM in KZN, serving an approximate population of eleven million [28]. Authors declared that they did not have access to patient information that could identify individual participants after data collection.

### Study design

This was a retrospective, descriptive study of patients with MM seen at the haematology clinic in KEH. The following data sets were collected from the patients' haematology clinic files and the laboratory information system (LIS); demographic data, presentation history (main complaint and duration of symptoms prior to reaching KEH), laboratory findings (albumin, $\beta_2M$, lactate dehydrogenase levels, FISH results, cytogenetics, haemoglobin level, renal function, calcium levels, bone marrow findings) radiology findings, records of first treatment regimen and response and duration of clinic follow up until the end of June 2022, to allow for a minimum of 2-year survival data capture. Patients were staged according to the MM International Staging System (ISS) rather than the Revised MM International staging system (R-ISS), as not all patients had molecular testing done.

For the molecular analysis, FISH probes used in the standard processing laboratory were grouped as follows: 13q14.3 deletion (D13S319), 17p13.1 /11q22.3 deletion *TP53/ATM*, 11q22 deletion *ATM* and t(4;14) *IGH@/FGFR3/WHSC1*. Other mutations detected by these standard probes were reported as additional mutations in the results.

Patients were assessed monthly; clinically and by serum protein electrophoresis with immuno-fixation. Treatment response after 6–8 cycles of chemotherapy was described as:

Complete response (CR): when the patient had single digit or no M-protein in serum (blood) and 24-hour urine, no bands on protein electrophoresis immunofixation, disappearance of any soft tissue plasmacytomas and $\leq$ 5% plasma cells in bone marrow.

Partial response (PR): when the patient had atleast a 50% reduction in M-protein and less than 5% plasma cells in the bone marrow with or without a reduction in 24-hour urinary M-protein by at least 90% (or less than 200 mg per 24 hours)

Stable disease was assigned when the patient had clinical improvement with improvement of the anaemia (transfusion independent) and recovery of renal function some response to treatment but with $<$ 25% improvement or progression in the M protein level [29]. Non-responders were defined as patients who did not achieve CR, PR, or stable disease after initiation of first-line therapy. These patients proceeded with second line therapies according to institutional policy.

A 24-month overall survival (OS) was calculated from time of diagnosis to follow-up at 2 years. The patients were confirmed alive through last recorded clinic visit. Patients not confirmed as deceased (through the national registration of deaths system) were grouped as those lost to follow-up (LTFU).

## Inclusion criteria

All new patients who were diagnosed as per World Health Organization Classification of Tumours of Haematopoietic and Lymphoid Tissues the 4th Edition (2008) and the Revised 4th Edition (2016) and treated for MM in KEH within the study period were included [30, 31].

## Exclusion criteria

Patients were excluded if diagnosed before or after the study period, diagnosed with other plasma cell dyscrasias other than MM, patients with suspected myeloma in whom criteria for diagnosis were not met and patients in whom HIV status was not recorded.

## Data collection

The haematology clinic new patient registers were sought, and the files of multiple myeloma patient diagnosed between July 2015 and June 2020 were selected and reviewed for data collection onto an excel spreadsheet. On the spreadsheet each included patient file was serially allocated a case identifying number (1–136) for purposes of statistical analysis. After completion of case file review and data collection the authors had no access to patient / case identifying data or information.

KEH uses a paper-based patient registry and thus the study period was informed by the availability of data. Additionally, a complete 60-month study period with a minimum of 2 years follow up period for each patient enrolled was sought.

The last three months of the study recruitment period was during the start of the COVID-19 pandemic in South Africa, this may have adversely affected recruitment.

## Data processing and statistical analysis

The statistical data analysis was conducted in R Statistical computing software of the R Core Team, 2020, version 3.6.3. The results were presented in the form of descriptive and inferential statistics. Where applicable, the descriptive statistics of numerical measurements were summarised as the minimum, maximum, quartiles, interquartile range, means, standard deviation and the coefficient of variation. The categorical variables were described as counts and percentage frequencies where simple and multiple bar charts were used to visually display the categorical variables. To determine the association between categorical variables, a Chi-Square Test was used, and when the distribution of the cross-tabulations contained an expected value of less than five, a Fisher's exact test was applied. All the inferential statistical analysis tests were conducted at 5% levels of significance.

## Ethical approval

Before data collection, the study was granted ethical approval by the local biomedical research ethics council, with protocol number BREC 00015662. As the study was retrospective, no specific study-related consent was obtained.

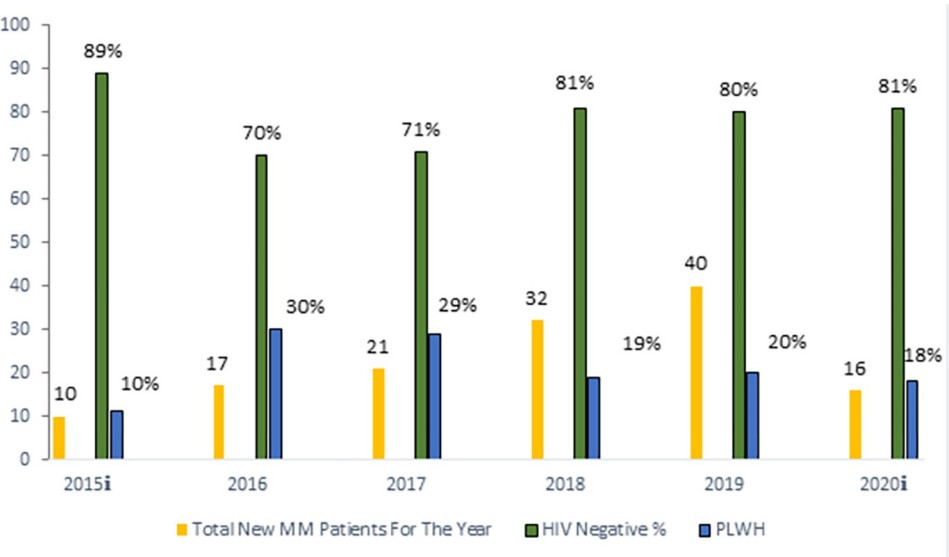

**Fig 1. The prevalence of HIV in multiple myeloma patients at KEH from July 2015 to June 2020.** Abbreviations: MM-multiple myeloma, KEH-King Edward VIII Hospital, KZN-KwaZulu-Natal, PLWH-People Living with HIV. Key: **i**–each comprise 6-months data.

## Results

The study cohort for the 5 years was 136 patients, with 29 patients being PLWH (21.3% CI 14.4–28.2). One patient without an HIV test result was excluded from the analysis.

The incidence of HIV in this MM cohort is 6 (six) cases/year (2016–2019).

The average new MM cases in the same timeframe were 27 cases/year (2016–2019) The prevalence of HIV in MM in the 60-month study period is 21.48%.

The incidence of HIV in MM is shown in Fig 1.

The patient profile (demographics including patient distance from KEH, clinico-pathological features) and disease stage are detailed in Tables 1 and 2. The study comprised of 85.9% patients (n = 116) of African ethnicity, 54.8% females (n = 74) and a M: F ratio of 1:1.25. The median age was 58 years, and the highest number of cases were recorded in the 51–70-year-olds (57.1%). In the age groups 40–50 years, the PLWH were significantly greater with 37.9% vs 14.4% respectively (p = 0.032).

### Molecular testing

A limited FISH MM panel was performed in 58.2% (n = 78) patients and cytogenetic karyotyping in 26.5% (n = 36). The panel included a total of 262 tests, that yielded 287 results (due to additional mutations detected with the standard FISH probes). Testing for deletion 11q22 yielded 10% additional results and that for t(4;14) yielded the most (20%) additional results. The results are described in Table 3.

A total of 33 patients were positive for the various FISH tests. The del 13q14.3 and del 11q22 were seen in 14 patients each. 17p13.1 was found in four patients and t (4;14) in one patient. HIV-negative patients were more likely to have an 11q22 mutation than PLWH (p = 0.027).

Additional mutations were detected in 17 patients yielding 25 positive results (eight in PLWH and 17 in HIV-negative patients) comprising Trisomy 11, 17p13.1/TP53 region duplications, 14q32/IGH region and FGFR3 region abnormalities. No statistically significant

**Table 1. Profile of multiple myeloma patients at King Edward VIII Hospital from 2015 to 2020.**

| HIV status | Negative | Positive | p-value | Overall¥ |
|---|---|---|---|---|
| | (N = 106) | (N = 29) | | (N = 135) |
| **Gender** | | | 0.642 | |
| Male | 49 (46.2%) | 12 (41.4%) | | 61 (45.2%) |
| Female | 57 (53.8%) | 17 (58.6%) | | 74 (54.8%) |
| **Race** | | | 0.098 | |
| African | 86 (81.9%) | 29 (100.0%) | | 115 (85.8%) |
| Indian | 11 (10.5%) | 0 (0.0%) | | 11 (8.2%) |
| White | 7 (6.7%) | 0 (0.0%) | | 7 (5.2%) |
| Mixed race^φ | 1 (1.0%) | 0 (0.0%) | | 1 (0.7%) |
| **Age group years** | | | 0.011 | |
| <40yrs | 7 (6.7%) | 3 (10.3%) | 1.000 | 10 (7.5%) |
| 40-50yrs | 15 (14.4%) | 11 (37.9%) | 0.032 | 26 (19.5%) |
| 51-70yrs | 62 (59.6%) | 14 (48.3%) | 1.000 | 76 (57.1%) |
| >70yrs | 20 (19.2%) | 1 (3.4%) | 0.176 | 21 (15.8%) |
| Mean±SD(CV%) | | | | 57.9±11.2(19.4) |
| Media (Q1-Q3) | | | | 58.0(50.0–65.0) |
| n(Min-Max) | | | | 135(29.0–84.0) |
| **Base hospital distance to KEH** | | | 0.776 | |
| <100km | 58 (58.0%) | 15 (55.6%) | | 73 (57.5%) |
| 100-150km | 10 (10.0%) | 4 (14.8%) | | 14 (11.0%) |
| >150km | 32 (32.0%) | 8 (29.6%) | | 40 (31.5%) |

¥ One patient with no traceable HIV results was excluded from the study

^φ Refers to people who identify as one born to parents or ancestors of different racial or ethnic backgrounds.

difference was noted between the two cohorts in the mutations associated with del 13q14.3, del 17p13.1 and t(4;14) (p values as noted in Table 3). Statistically significant differences were noted in the mutations associated with del 11q22 / chromosome 11 (p = 0.027). However, the row-pairwise tests could not be conducted due to smaller counts that could lead to Type-I error.

## Treatment

The different first-line therapeutic modalities are depicted in Table 4. 117 patients (86.7%) received treatment. 18 patients (13.3%) did not follow-up long enough to start any treatment regimen. The therapeutic options were chemotherapy alone or in various combinations with radiotherapy stem cell transplant, surgery, and haemodialysis. Combination chemotherapy comprised cyclophosphamide, thalidomide, dexamethasone with Zoledronic acid (CTD-Z), with or without radiotherapy, surgery, autologous stem cell transplant and haemodialysis. If CTD-Z was used, the combination chemotherapy and/or radiotherapy were followed by maintenance therapy with thalidomide upon confirmation of complete response or partial response [29, 32]. Other treatment combinations were; vincristine, doxorubicin, dexamethasone (VAD) in a small minority of patients with renal failure and melphalan with prednisone (MP). Five younger patients were referred for ASCT after 8 cycles of VAD. 58% of patients showed a response to first-line treatment and remained only on maintenance therapy with single-agent thalidomide.

**Table 2. Symptoms and clinical presentation of multiple myeloma patients treated at King Edward VIII Hospital from 2015 to 2020.**

| HIV status | Negative | Positive | p-value | Overall |
|---|---|---|---|---|
| | (N = 106) | (N = 29) | | (N = 135) |
| **Presenting complaints*** | | | 0.508 | |
| Fractures | 45 (46.9%) | 12 (48.0%) | | 57 (47.1%) |
| Bone pain | 36 (37.5%) | 6 (24.0%) | | 42 (34.7%) |
| Neuropathy | 28 (29.2%) | 7 (28.0%) | | 35 (28.9%) |
| Renal failure | 16 (16.7%) | 3 (12.0%) | | 19 (15.7%) |
| Anaemia | 8 (8.3%) | 5 (20.0%) | | 13 (10.7%) |
| Hypercalcemia | 11 (11.5%) | 2 (8.0%) | | 13 (10.7%) |
| Other | 12 (12.5%) | 6 (24.0%) | | 18 (14.8%) |
| **Symptom duration groups in months** | | | 0.634 | |
| <3 | 12 (15.8%) | 3 (18.8%) | | 15 (16.3%) |
| 3-<6 | 26 (34.2%) | 3 (18.8%) | | 29 (31.5%) |
| 6-<12 | 25 (32.9%) | 6 (37.5%) | | 31 (33.7%) |
| 12+ | 13 (17.1%) | 4 (25.0%) | | 17 (18.5%) |
| **ISS stage** | | | 0.200 | |
| Stage I | 21 (26.3%) | 2 (10.0%) | | 23 (23.0%) |
| Stage II | 23 (28.8%) | 5 (25.0%) | | 28 (28.0%) |
| Stage III | 36 (45.0%) | 13 (65.0%) | | 49 (49.0%) |
| **MDEs** | | | | |
| **Anaemia** | | | 0.103 | |
| Absent | 39 (36.8%) | 6 (20.7%) | | 45 (33.3%) |
| Present | 67 (63.2%) | 23 (79.3%) | | 90 (66.7%) |
| **Renal failure** | | | 0.489 | |
| Absent | 73 (68.9%) | 18 (62.1%) | | 91 (67.4%) |
| Present | 33 (31.1%) | 11 (37.9%) | | 44 (32.6%) |
| **Hypercalcaemia** | | | 0.111 | |
| Absent | 96 (90.6%) | 23 (79.3%) | | 119 (88.1%) |
| Present | 10 (9.4%) | 6 (20.7%) | | 16 (11.9%) |

Abbreviations: ISS-International Staging System

*Patients presented with multiple complaints; hence the number of complaints is > 135

All patients that responded to chemotherapy, proceeded to maintenance therapy with Thalidomide. Patients that were eligible and agreeable, were referred for autologous stem cell transplantation. At the time of this study, no patients received Bortezomib or other novel treatments as these drugs were not available in the public sector in KZN.

Treatment and outcomes were analysed and compared between the two cohorts as seen in Table 5. 86.7% (n = 117) of patients received treatment. Treatment response was analysable in 69% of these patients (n = 81). Approximately 58% (n = 47) showed a response (achieving a CR, PR, or stable disease). There were no statistically significant differences in response between the PLWH and HIV-negative groups (p = 0.926).

% And p-values based on non-missing cases | * parametric p-value comparing HIV-negative and PLWH MM groups

Outcomes and survival at 24 months were calculated as depicted in Table 6. A median follow-up period of 18 months was recorded for both HIV-positive and negative MM groups. Survival ranged from one day to 84 months. The survival outcomes for the two cohorts are detailed in Table 6.

**Table 3. Genetic mutations in multiple myeloma patients treated at King Edward VIII Hospital.**

| HIV status | Negative | Positive | p-value* | Overall |
|---|---|---|---|---|
| **Cytogenetic Karyotyping** | (n = 30) | (n = 6) | 0.535 | (n = 36) |
| Normal | 27 (90.0%) | 5 (83.3%) | | 32 (88.9%) |
| Abnormal | 3 (10.0%) | 1 (16.7%) | | 4 (11.1%) |
| **FISH** | | | | |
| **13q14.3 deletion (D13S319)** | (n = 60) | (n = 14) | 0.291 | (n = 74) |
| Negative | 48 (80.0%) | 11 (78.6%) | | 59 (79.7%) |
| Positive | 12 (20.0%) | 2 (14.3%) | | 14 (18.9%) |
| Additional mutations~ | 00 (0.0%) | 1 (7.1%) | | 1 (1.4%) |
| **17p13.1 /11q22.3 deletion *TP53/ATM*** | (n = 61) | (n = 14) | 0.318 | (n = 75) |
| Negative | 54 (88.5%) | 11 (78.6%) | | 65 (86.7%) |
| Positive | 3 (4.9%) | 1 (7.1%) | | 4 (5.3%) |
| Additional mutations~ | 4 (6.6%) | 2 (14.3%) | | 6 (8.0%) |
| **11q22 deletion *ATM*** | (n = 52) | (n = 13) | 0.027# | (n = 65) |
| Negative | 35 (67.3%) | 10 (76.9%) | | 45 (69.2%) |
| Positive | 14 (26.9%) | 0 (0.0%) | | 14 (21.5%) |
| Additional mutations~ | 3 (5.8%) | 3 (23.1%) | | 6 (9.2%) |
| **T (4;14) *IGH@/FGFR3/WHSC1*** | (n = 59) | (n = 14) | 0.278 | (n = 73) |
| Negative | 49 (83.1%) | 11 (78.6%) | | 60 (82.2%) |
| Positive | 0 (0.0%) | 1 (7.1%) | | 1 (1.4%) |
| Additional mutations~ | 10 (16.9%) | 2 (14.3%) | | 12 (16.4%) |

* Parametric p-value refers to comparison between PLWH and the HIV- groups.

~ Additional mutations were reflex tested owing to preparation of probes used for the standard mutations sought.

# The row pairwise tests could not be conducted due to smaller counts that were likely to lead to Type I error.

## Discussion

In this study, we describe the clinico-pathological profile and survival of MM patients in KZN, South Africa, and compare MM in PLWH with the HIV-negative cohort. We recorded a steady increase in the incidence MM in 2016–2019 and the lower numbers in 2020 are most

**Table 4. Treatment options received by MM patients in KEH VIII (2015–2020).**

| HIV status | Negative (N = 91) | Positive (N = 26) | Overall (N = 117) |
|---|---|---|---|
| **Treatment** | | | |
| Chemotherapy alone | 49 (53.8%) | 14 (53.8%) | 63 (53.8%) |
| Chemotherapy; Hemodialysis; Radiotherapy | 1 (1.1%) | 0 (0.0%) | 1 (0.9%) |
| Chemotherapy; Stem cell transplant | 1 (1.1%) | 0 (0.0%) | 1 (0.9%) |
| Chemotherapy; Surgery | 3 (3.3%) | 0 (0.0%) | 3 (2.6%) |
| Chemotherapy; Radiotherapy; Hemodialysis | 1 (1.1%) | 0 (0.0%) | 1 (0.9%) |
| Chemotherapy; Radiotherapy | 10 (11.0%) | 2 (7.7%) | 12 (10.3%) |
| Chemotherapy; Radiotherapy; Stem cell transplant | 1 (1.1%) | 0 (0.0%) | 1 (0.9%) |
| Chemotherapy; Stem cell transplant | 2 (2.2%) | 0 (0.0%) | 2 (1.7%) |
| Surgery | 1 (1.1%) | 0 (0.0%) | 1 (0.9%) |
| Radiotherapy; Chemotherapy | 20 (22.0%) | 9 (34.6%) | 29 (24.8%) |
| Radiotherapy; Chemotherapy; Stem cell transplant | 1 (1.1%) | 0 (0.0%) | 1 (0.9%) |
| Radiotherapy; Chemotherapy; Surgery | 1 (1.1%) | 0 (0.0%) | 1 (0.9%) |
| Radiotherapy | 0 (0.0%) | 1 (3.8%) | 1 (0.9%) |

**Table 5. Treatment and outcomes in MM patients in KEH VIII Hospital.**

| HIV status | Negative | Positive | p-value* | Overall |
|---|---|---|---|---|
| | (N = 106) | (N = 29) | | (N = 135) |
| **Received treatment** | | | 0.763 | |
| No | 15 (14.2%) | 3 (10.3%) | | 18 (13.3%) |
| Yes | 91 (85.8%) | 26 (89.7%) | | 117 (86.7%) |
| **Treatment outcome** | n = 68 | n = 13 | 0.926 | n = 81 |
| Response | 39 (57.4%) | 8 (61.5%) | | 47 (58.0%) |
| No response | 29 (42.7%) | 5 (48.5%) | | 34 (42%) |
| **Clinic follow-up duration (months)** | | | 0.778 | |
| Median(Q1-Q3) | 18.1(2.78–33.3) | 18.0(1.69–33.8 | | 18.0(2.69–33.6) |
| n(Min-Max) | 106(0.0661–75.0) | 29(0.264–63.7) | | 135(0.0661–75.0) |

likely attributable to the shorter study period and the SARS-Cov2 lockdown in South Africa that was from March 2020.

Contrary to our study findings, MM is globally known to predominantly affect African males [5, 7, 9, 14, 33, 34]. In 2009, another South African study by Rankapole et al, found a similarly higher proportion of females with MM and the South African National Institute of Communicable Diseases National Cancer Registry, also recorded an increasing incidence of MM in African females from 2016–2018 [6, 14]. In 2016 Van der Walt reported an equivalent gender distribution in Gauteng, South Africa. The gender distribution in these two provinces of South Africa is therefore reflective of the population statistics. [4, 6, 26, 27, 35–40].

Mhlanga et al in Gauteng, noted that in first-world countries, access to health care is influenced by various factors that account for women accessing health care more often than men; including susceptibility to infections and chronic diseases owing to childminding and need for contraception and childbirth. In South Africa, a further imbalance exists whereby resources are limited, and individual finances limit access to health services, with 71.5% of households utilising public health services. Individuals in female-headed households were more likely to access health earlier than male-headed households [41]. An Eastern Cape study in South Africa reported that men need to provide for their families and with an inflexible health system, this led them to delay accessing healthcare or to default their follow-up and treatment [42]. These additional socioeconomic factors were also likely to contribute to the larger number of females in our cohort.

The median age in our study was 58 years, and the highest was seen in the 51–70-year-olds. In the age groups 40–50 years the PLWH were proportionately significantly more, a finding supported by another South African study [18].

**Table 6. Survival for multiple myeloma patients treated in King Edward VIII Hospital over 2 years.**

| HIV status | Negative (N = 106) | Positive (N = 29) | p-value | Overall (N = 135) |
|---|---|---|---|---|
| **Year 1** | | | 0.384 | |
| Alive | 61 (57.5%) | 13 (44.8%) | | 74 (54.8%) |
| Died | 5 (4.7%) | 1 (3.4%) | | 6 (4.4%) |
| Lost to follow-up | 40 (37.7%) | 15 (51.7%) | | 55 (40.7%) |
| **Year 2** | | | 0.792 | |
| Alive | 45 (42.5%) | 12 (41.4%) | | 57 (42.2%) |
| Died | 7 (6.6%) | 1 (3.4%) | | 8 (5.9%) |
| Lost to follow-up | 54 (50.9%) | 16 (55.2%) | | 70 (51.9%) |

Most patients presented to our treating centre with a history of 6–12 months symptomatology and approximately one-fifth had symptoms for more than a year. Late presentation to care is well documented with regards to HIV care.

In Canada, it is reported that approximately 70–80% of patients diagnosed, during hospitalisation, with either HIV disease or other diseases managed to follow-up back to hospital within 3 months [43]. In contrast, other Canadian data showed that late presentation and late diagnosis (defined as a duration greater than 3 months from diagnosis to presentation to care) were noted in approximately 50% of patients, similar to other high-income countries. This late presentation was associated with African descent and surprisingly, female gender [43–46]. In other first-world countries, in an Embase and Medline meta-analysis, an interval of 163 days from first symptoms to diagnosis in MM is reported [47]. A UK electronic records data analysis reported that bone pain was recorded, on average 7 months prior to the diagnosis of MM [48].

Whilst developed countries also have challenges of late presentation and late diagnosis [46], our patients presented to our unit even later, with a history of the presenting complaint generally longer than 6 months, despite the fact that the majority resided within a 100km radius of the haematology unit. A similar late presentation has been described in Ghana where patients presented to hospital one year after onset of symptoms. However, this did not impact survival [7]. Although in our analysis PLWH presented with a shorter duration of symptoms, this was not significantly different to the HIV-negative population and their outcomes were comparable.

The late presentation in our study cohort could be due to multiple social factors and financial constraints, and delays in patient diagnosis from the outlying hospitals [4, 13]. The low index of suspicion for MM among primary care clinicians results in the patients "initial symptoms" being ignored or missed [13]. This is evident in the prolonged time to diagnosis that appears to cross the socio-economic divide and the recurring sentiment to put effort into shortening time to presentation and diagnosis in MM [47–50]. Whilst a week to diagnosis is impossible, future prospective studies may assist in deciding on the most reasonable duration to presentation and more importantly, how to achieve a quicker diagnostic algorithm.

Anaemia is reportedly the most common presenting complaint in MM, and this concurred with our laboratory findings [7, 33, 48]. However, patients predominantly complained of bone pain rather than symptoms of anaemia, upon presentation to hospital. A comparative study of 110 patients in Israel, found that bone pain was present for up to 2 years in MM patients prior to diagnosis compared to non-MM patients. They suggest that back pain accompanied by fatigue, weight loss or abnormal results should raise a "red flag" for possible MM [51].

Our findings showed a tendency towards more MM, in PLWH, presenting with anaemia. Although this was not statistically significant, anaemia is a known complication of HIV infection and treatment. The aetiology of anaemia in HIV disease is multi-factorial; chronic disease state, anti-retroviral therapy, haemolysis, other malignancies, nutritional deficiencies and HIV associated renal dysfunction [17, 48, 52].

However, anaemia is also a diagnostic criterion for MM and MM is associated with renal impairment [53, 54]. The challenge in treating anaemia, diagnosing MM, and managing HIV disease in this population lies therefore in establishing the aetiology of the anaemia.

Additionally, anaemia that is not secondary to MM in PLWH and suffering with MM, can limit therapeutic options and chemotherapy outcomes.

In our study, a larger group presented with pathological fractures, similar to three other studies Ghana (44% of the study cohort had fractures), Uganda (85.3% of patients had skeletal pathology) and Indonesia found that bone pain and osteolytic lesions constituted 76.9% of their study population [5, 7, 34]. Supporting our study findings, data from another province in

South Africa found that pathological fractures were comparable in PLWH and HIV-negative MM patients [18]. Further studies are important in this case to describe the long-term outcome of this type of presentation, and to consider pathological fractures for addition to the diagnostic criteria for MM. Increased awareness and a high index of suspicion when investigating a patient with anaemia who also complains of bone pain or presents with pathological fractures could lead to an earlier diagnosis of MM in our setting, [47, 53].

In our study, approximately a third of our patients had renal impairment meeting the criteria for MM diagnosis, although hypercalcaemia was not a prominent feature [31]. Proportionately the PLWH had a slightly higher prevalence of renal impairment. This difference was not statistically significant, and this contrasts with another South African study where PLWH had less renal impairment [18]. Other African studies also found almost a third of the patients presented with renal impairment and hypercalcemia [4, 7]. An Asian study also found approximately 38,5% of patients with increased creatinine levels, but only 4.2% patients with hypercalcaemia [34]. Hypercalcemia can be fatal in the acute setting and in resource-poor settings, renal impairment is an impediment to treatment of hypercalcemia. Further investigation on effective control or prevention of hypercalcaemia in the acute setting in the absence of haemodialysis could contribute to better outcomes in MM.

Haemodialysis is an important support for MM patients as renal recovery is associated with improved survival in MM. Haemodialysis together with timeous chemotherapy in the Acute kidney injury-setting may prove complementary, as with disease remission and suppression of the inflammatory process, patients can develop independence from dialysis [55–57]. However, in our study cohort, <10% received haemodialysis as per South African guidelines [54]. Haemodialysis in South Africa is still a scarce resource that requires investment. Careful revision and consideration of treatment guidelines for haemodialysis in MM could also improve patient outcomes.

Our study found an HIV prevalence in MM to be similar to that described for HIV in KZN 21.48 vs 19.1% [15]. Although some studies report an increased risk and incidence of MM in HIV, a causative relationship between HIV and MM has not been established. Most publications are case reports on HIV and MM [58, 59]. The demographics, presentation, laboratory findings, outcomes and survival in the HIV-negative group and PLWH groups were also comparable, corroborated by one other South African study [18].

Molecular testing was under-utilised for purposes of staging and risk stratification in our cohort as not all patients had FISH analysis at the time of diagnostic bone marrow biopsy. Del 11q22 and del13q14.3 were seen mostly in HIV- negative patients. Very few patients were positive for del 17p13.1, which is one of the markers of high-risk disease. Additional genetic abnormalities detected during standard FISH analysis, involving Trisomy 11 are suggestive of hyperdiploid MM, which has good prognosis [25]. Prospective studies are warranted on molecular abnormalities in PLWH with MM and on suitable therapy targeting both MM and HIV which might yield improved survival and data in PLWH. This is particularly pertinent with the incorporation of molecular testing in the newly published classification of MM by the International Consensus Criteria [60].

In the literature search conducted, the prevalence of 11q22 mutations in HIV positive MM patients as well as its significance were not described.

Owing to limitation of local FISH probe design, it is not possible to determine whether the additional mutations detected with del11q22 testing are associated with the Ataxia-telangiectasia mutated (ATM) gene or other subset mutations.

Approximately half of our patients presented with ISS Stage III disease. This is a common phenomenon, with a proportionately greater number of patients presenting with advanced disease in most studies worldwide [25, 61]. Stage III disease is independently associated with

decreased overall survival [5, 26, 34, 62, 63]. It is possible that this late presentation in developing countries is secondary to delays in diagnosis. In South Africa, patient-related factors are also likely to contribute to the delayed presentation.

Our 2-year OS of 42.2% compares well with the one-year OS of 51,6% in a Ghanaian study. However, our median survival of only 18 months is inferior to the median survival of 53 months in Ghana, despite late presentation in one-third of patients [7]. Studies to confirm improved overall survival where complete remission is achieved in the first year of therapy and those that follow patients for longer periods suggest that survival of up to ten years is achievable [58, 64, 65]. Factors impacting prognosis and outcome in local patients are worthy of further prospective study.

## Limitations

Whilst incomplete investigation was a drawback in this study, where only 101 patients (74.3%) had complete staging and only 78 (58.2%) had molecular testing), the authors have analysed the records within the constraints of what best could be retrieved. Owing to the paucity of results, comparative analysis for some FISH tests was hampered as this posed a risk of type 1 statistical errors. Since the numbers for the other parameters were large enough for statistical analysis, it is our impression that the poor record-keeping did not affect the analysis significantly.

The COVID-19 pandemic may have adversely affected the study since approximately the last 10% of the study period was during a hard lockdown in South Africa. This may have delayed referrals of new patients and interrupted the treatment of known patients further skewing our findings.

## Conclusion

This study described a single-centre experience with a large number of MM patients, presenting predominantly with late-stage disease and poor survival outcomes. There was no difference in survival outcomes in PLWH compared with the HIV-negative MM cohort. Earlier diagnosis, and better therapeutic options are required in the public sector in South Africa. In a population of 11 million in the province of KZN, such high prevalence of disease and advanced stage of disease despite proximity to the referring hospital warrants attention and improved intervention from government and public stakeholders to create awareness, improve care and better outcomes.

Based on our study findings, the implementation of international guidelines as standard of care is recommended, within the economic constraints of laboratory testing and available therapeutics.

The following recommendations in the local setting in KZN are therefore suggested.:

- Early referral and diagnosis including early involvement of the haematologist.

- Bone pain and anaemia should raise suspicion for MM in our setting.

- Mandatory HIV testing in MM.

- Full risk stratification at diagnosis in order to motivate for novel therapies.

- Intentional patient classification at the clinic level to identify those eligible for ASCT.

- Investment in improved record keeping and further collection of evidence.

- Further prospective study on the use of PI antiretroviral therapy in PLWH with MM.

Following this study, some of these measures, including awareness and education, a diagnostic algorithm and social intervention are being investigated in a prospective study locally.

## Supporting information

**S1 Data.**
(XLSX)

## Acknowledgments

Professor Irene. Mackraj and Dr Nadine. Rapiti
Dr Usri Ibrahim for technical assistance.

## Author Contributions

**Conceptualization:** Lungisile Hildegard Chili, Irene Mackraj, Nadine Rapiti.

**Data curation:** Lungisile Hildegard Chili.

**Formal analysis:** Lungisile Hildegard Chili, Nadine Rapiti.

**Investigation:** Lungisile Hildegard Chili.

**Methodology:** Lungisile Hildegard Chili, Irene Mackraj, Nadine Rapiti.

**Project administration:** Lungisile Hildegard Chili.

**Supervision:** Irene Mackraj, Nadine Rapiti.

**Validation:** Irene Mackraj, Nadine Rapiti.

**Visualization:** Lungisile Hildegard Chili, Irene Mackraj, Nadine Rapiti.

**Writing – original draft:** Lungisile Hildegard Chili.

**Writing – review & editing:** Lungisile Hildegard Chili, Irene Mackraj, Nadine Rapiti.

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
