## [Decision Letter · Decision Letter 0]

2 May 2023

PONE-D-23-09091Profile and Outcome of Multiple Myeloma with and without HIV treated at a Tertiary Hospital in KwaZulu-Natal, South AfricaPLOS ONE

Dear Dr. Chili

Thank you for submitting your manuscript to PLOS ONE. After careful consideration, we feel that it has merit but does not fully meet PLOS ONE’s publication criteria as it currently stands. Therefore, we invite you to submit a revised version of the manuscript that addresses the points raised during the review process.

We look forward to receiving your revised manuscript.

Kind regards,

Zivanai Cuthbert Chapanduka, MBChB (M.D)

Academic Editor

PLOS ONE

Journal Requirements:

Reviewers' comments:

Reviewer's Responses to Questions

**Comments to the Author**

1. Is the manuscript technically sound, and do the data support the conclusions?

Reviewer #1: Partly

Reviewer #2: Yes

2. Has the statistical analysis been performed appropriately and rigorously? 

Reviewer #1: Yes

Reviewer #2: Yes

3. Have the authors made all data underlying the findings in their manuscript fully available?

Reviewer #1: Yes

Reviewer #2: Yes

4. Is the manuscript presented in an intelligible fashion and written in standard English?

Reviewer #1: Yes

Reviewer #2: Yes

5. Review Comments to the Author

Reviewer #1: Dear Authors,

Thank you for submitting your research for publication. This is an interesting topic and study. However, there are areas that need clarification and improvement.

General comments:

- Would you please clarify how data was collected when authors declared that they had no access to patient’s information?

- The term incidence and prevalence were used interchangeably in this manuscript which creates confusion.

o Does figure -1 show incidence or prevalence of MM?

o Is the calculated prevalence figure-1 from the total KZN population?

o What do you mean by incidence of HIV in MM, is it frequency or incidence?

- In Table-1: what do you mean by coloured race? Is it mixed race? Correct typing error “Media”

- Please refer to table-2 in the result section.

- In the result section the incidence/prevalence reported as years i.e., 2015, 2016, etc. However, the study started July 2015 (6 months only in 2015) and ended June 2020 (again 6 months only in 2020). Does it mean the calculated incidence/prevalence in 2015 was obtained from 6 months data compared to 12 months in 2016? This part of the study is vague and needs clarification. Is this the reason for fewer patients in 2015 and 2020 in fig-1?

- Please reference each sentence/fact.

- There are several typo-grammatical errors. I highlighted some in the PDF, please correct these.

- Please include line numbers in the manuscript file. Use continuous line numbers (do not restart the numbering on each page). As per PLOS ONE guidelines.

- Both American and British English were used in this manuscript. Please be consistent.

- Referencing errors. Please use Vancouver referencing style.

Abstract:

Conclusion: “MM prevalence in KZN is increasing” Increasing compared to what? The result section in the abstract did not show increased MM cases.

Introduction:

“It has also been reported that some ART protease inhibitors (PIs) are effective in MM treatment, functioning similarly to the Proteosome Inhibitor therapy known to be effective in MM.” please reference this sentence.

Method:

- “ Authors declared that they did not have access to patient information that could identify individual participants during and after data collection.”. Please clarify method of data collection method when investigators had no access to patients’ information.

- “All patients that responded to chemotherapy, proceeded to maintenance therapy with Thalidomide. Patients that were eligible and agreeable, were referred for autologous stem cell transplantation. At the time of this study, no patients were on Bortezomib or other novel treatments, as these drugs were not available in the public sector in KZN.”. Consider moving these sentences to the result section.

Result:

- “The prevalence for MM treated at KEH over the 5-year study period and the incidence of HIV in MM are shown in Figure 1.”. As mentioned above the term prevalence and incidence were used interchangeably creating confusion, pleas A limited FISH MM panel was performed in 58.2% (n=78) patients and cytogenetic

- “ karyotyping in 26.5% (n=36). The panel included a total of 262 tests, that yielded the following 287 results (due to additional mutations detected with the standard FISH probes): deletion 13q14.3 (73 tests/ 74 results), deletion 17p13.1 (69 tests / 75 results), deletion 11q22 (59 tests /65 results) and t (4;14) (61 tests / 73 results).”. Please clarify this sentence. Consider rephrasing.

Discussion:

- “Further studies are important in this case to differentiate the long-term outcome of this type of presentation, and to consider this bone pain and fractures as possible inclusion criteria in the diagnostic algorithm (rather than lytic lesions alone). “ . fractures is a late finding. Is it right to consider including fractures rather than lytic lesions in the diagnostic criteria of myeloma?

- “Administration of haemodialysis with chemotherapy is complementary, as with disease response to chemotherapy, patients can develop independence from dialysis.”. Please rephrase

Best wishes

Reviewer #2: A well written manuscript with good sound statistical analysis.

Some additional points to consider:

While there is no significant statistical difference in anaemia at presentation between HIV positive and negative groups, there is certainly a tendency to have more anaemia in the HIV positive cohort. Anaemia is a well known complication of HIV infection and its treatment. Comment on how this may complicate MM diagnosis and management is paramount.

11q22 is significantly higher in HIV positive patients. A comment on the possible reasons for this, if described in literature, or just stating not described is important as this is one of the major findings.

6. PLOS authors have the option to publish the peer review history of their article (what does this mean?). If published, this will include your full peer review and any attached files.

Reviewer #1: No

Reviewer #2: **Yes: **Leonard Mutema

---

## [Author Response · Author response to Decision Letter 0]

22 May 2023

CORRECTIONS:

Reviewer 1: Comments and author responses

1.

•Would you please clarify how data was collected when authors declared that they had no access to patient’s information?

•AND POINT 13 BELOW:

Method:

“ Authors declared that they did not have access to patient information that could identify individual participants during and after data collection.”. Please clarify method of data collection method when investigators had no access to patients’ information.

This was clarified in the method section and written as quoted below:

“Data collection

The haematology clinic new patient registers were sought, and the files of multiple myeloma patient diagnosed between July 2015 and June 2020 were selected and reviewed for data collection onto an excel spreadsheet. On the spreadsheet each included patient file was serially allocated a case identifying number (1-136) for purposes of statistical analysis. After completion of case file review and data collection the authors had no access to patient / case identifying data or information.

KEH uses a paper-based patient registry and thus the study period was informed by the availability of data. Additionally, a complete 60-month study period with a minimum of 2 years follow up period for each patient enrolled was sought.

The last three months of the study recruitment period was during the start of the COVID-19 pandemic in South Africa, this may have adversely affected recruitment.”

2. The term incidence and prevalence were used interchangeably in this manuscript which creates confusion.

1. Does figure -1 show incidence or prevalence of MM?

2. Is the calculated prevalence figure-1 from the total KZN population?

3. What do you mean by incidence of HIV in MM, is it frequency or incidence?

• Figure 1 shows recorded new cases of MM within which the prevalence of HIV is reported 21.48%. 

The incidence of HIV in our MM cohort is only comparable for the 4 years 2016- 2019 and is 6 cases/year (2016-2019)

Whereas the incidence of MM in the same time frame was 27 cases/year (2016 – 2019)

The prevalence of HIV in MM is thus noted in the results and discussion.

• The calculated prevalence refers to the state dependent population of KZN. This is noted in the limitations as we have no access to the records of private care users.

• The use of the terms prevalence (which means frequency) and incidence, calculated per time period are corrected. 

3. In Table-1: what do you mean by coloured race? Is it mixed race? Correct typing error “Media”

The term “Mixed race” is now used and described below. This new term means exactly what I was trying to say using the term Coloured race.

Refers to people who identify as one born to parents or ancestors of different racial or ethnic backgrounds.

4. Please refer to table-2 in the result section.

In the result section the incidence/prevalence reported as years i.e., 2015, 2016, etc. However, the study started July 2015 (6 months only in 2015) and ended June 2020 (again 6 months only in 2020). Does it mean the calculated incidence/prevalence in 2015 was obtained from 6 months data compared to 12 months in 2016? This part of the study is vague and needs clarification. Is this the reason for fewer patients in 2015 and 2020 in fig-1?

- The comparison is between HIV + and HIV negative patients in the 60-month period

- Prevalence rather than incidence is used and corrected in the whole text, changes are tracked.

- Annual comparisons have been corrected in the results text with comparison made and incidence or prevalence calculated appropriately with reference to 2016-2019 time periods eg. the average new cases of MM were 27/year and that of new MM with HIV were 6/year.

- The prevalence of HIV in MM could be calculated for the full 5 year / 60-month period and was (29/135 = 21.48%). This is the calculation that is compared to the KZN HIV prevalence statistics. Please note that the HIV status of one patient could not be confirmed and this patient was excluded from some calculations.

Statements now included in text are:

“The incidence of HIV in this MM cohort is 6 cases/year (2016-2019).

The average new MM cases in the same time frame was 27 cases/year (2016 – 2019) The prevalence of HIV in MM in the 60-month study period is 21.48%. 

The incidence of HIV in MM is shown in Fig 1.”

5. Please reference each sentence/fact. 

Done.

6. There are several typo-grammatical errors. I highlighted some in the PDF, please correct these.

Done – used word editor too.

7. Please include line numbers in the manuscript file. Use continuous line numbers (do not restart the numbering on each page). As per PLOS ONE guidelines.

Done.

8. Both American and British English were used in this manuscript. Please be consistent.

English (UK) now used.

9. Referencing errors. Please use Vancouver referencing style.

Vancouver referencing style used.

10. Abstract:

Conclusion: “MM prevalence in KZN is increasing” Increasing compared to what? The result section in the abstract did not show increased MM cases.

Corrected and rephrased to read: 

“The incidence of HIV in newly diagnosed MM patients in KZN was increasing.” 

11. Introduction:

“It has also been reported that some ART protease inhibitors (PIs) are effective in MM treatment, functioning similarly to the Proteosome Inhibitor therapy known to be effective in MM.” please reference this sentence.

Done.

Mendez-Lopez M, Sutter T, Driessen C, Besse L. HIV protease inhibitors for the treatment of multiple myeloma. Clin Adv Hematol Oncol. 2019;17(11):615-23.

12. Method:

“Authors declared that they did not have access to patient information that could identify individual participants during and after data collection.”. Please clarify method of data collection method when investigators had no access to patients’ information.

Clarified, please see point number 1 above. 

13. “All patients that responded to chemotherapy, proceeded to maintenance therapy with Thalidomide. Patients that were eligible and agreeable, were referred for autologous stem cell transplantation. At the time of this study, no patients were on Bortezomib or other novel treatments, as these drugs were not available in the public sector in KZN.”. Consider moving these sentences to the result section.

Moved appropriately to results section. And rephrased slightly. 

“All patients that responded to chemotherapy, proceeded to maintenance therapy with Thalidomide. Patients that were eligible and agreeable, were referred for autologous stem cell transplantation. At the time of this study, no patients received Bortezomib or other novel treatments as these drugs were not available in the public sector in KZN.”

14 Result:

“The prevalence for MM treated at KEH over the 5-year study period and the incidence of HIV in MM are shown in Figure 1.”. As mentioned above the term prevalence and incidence were used interchangeably creating confusion, pleas…

Figure 1 was corrected to state:

The incidence of HIV in MM. The context population are the new MM patients.

Relevant symbols applied to better describe time frames.

Time period comparisons with reference to this figure were clarified. 

Statements in the manuscript now read:

“The incidence of HIV in this MM cohort is 6 cases/year (2016-2019).

The average new MM cases in the same time-frame was 27 cases/year (2016 – 2019) The prevalence of HIV in MM in the 60-month study period is 21.48%. 

The incidence of HIV in MM is shown in Figure 1.”

15 A limited FISH MM panel was performed in 58.2% (n=78) patients and cytogenetic some Rephrase karyotyping in 26.5% (n=36). The panel included a total of 262 tests, that yielded the following 287 results (due to additional mutations detected with the standard FISH probes): deletion 13q14.3 (73 tests/ 74 results), deletion 17p13.1 (69 tests / 75 results), deletion 11q22 (59 tests /65 results) and t (4;14) (61 tests / 73 results).”. Please clarify this sentence. Consider rephrasing.

The statement was recalled and replaced with the following statements:

” A limited FISH MM panel was performed in 58.2% (n=78) patients and cytogenetic karyotyping in 26.5% (n=36). The panel included a total of 262 tests, that yielded 287 results (due to additional mutations detected with the standard FISH probes). Testing for deletion 11q22 yielded 10% additional results and that for t(4;14) yielded the most (20% )additional results. The results are described in Table 3.”

16 Discussion:

Further studies are important in this case to differentiate the long-term outcome of this type of presentation, and to consider this bone pain and fractures as possible inclusion criteria in the diagnostic algorithm (rather than lytic lesions alone). “ . fractures is a late finding. Is it right to consider including fractures rather than lytic lesions in the diagnostic criteria of myeloma? Certainly not. The sentence rephrased.

The consideration is to add pathological fractures to the diagnostic criteria.

Statement now in the manuscript reads:

‘Further studies are important in this case to describe the long-term outcome of this type of presentation, and to consider pathological fractures for addition to the diagnostic criteria for MM.”

17 

Administration of haemodialysis with chemotherapy is complementary, as with disease response to chemotherapy, patients can develop independence from dialysis.”. Please rephrase

Rephrased to:

“Haemodialysis together with timeous chemotherapy in the Acute kidney injury-setting may prove complementary, as with disease remission and suppression of the inflammatory process, patients can develop independence from dialysis.”

Reviewer 2 : Comments and author responses

18 Some additional points to consider:

While there is no significant statistical difference in anaemia at presentation between HIV positive and negative groups, there is certainly a tendency to have more anaemia in the HIV positive cohort. Anaemia is a well known complication of HIV infection and its treatment. 

Comment on how this may complicate MM diagnosis and management is paramount.

This point was considered and captured in the discussion thus:

“Our findings showed a tendency towards more MM, in PLWH, presenting with anaemia. Although this was not statistically significant, anaemia is a known complication of HIV infection and treatment. The aetiology of anaemia in HIV disease is multi-factorial; chronic disease state, anti-retroviral therapy, haemolysis, other malignancies, nutritional deficiencies and HIV associated renal dysfunction[17, 48, 52]. 

However, anaemia is also a diagnostic criterion for MM and MM is associated with renal impairment [53, 54]. The challenge in treating anaemia, diagnosing MM, and managing HIV disease in this population lies therefore in establishing the aetiology of the anaemia.

Additionally, anaemia that is not secondary to MM in PLWH and suffering with MM , can limit therapeutic options and chemotherapy outcomes.”

19 11q22 is significantly higher in HIV positive patients. A comment on the possible reasons for this, if described in literature, or just stating not described is important as this is one of the major findings.

 A comment was added in the discussion regarding this suggestion:

‘In the literature search conducted, the prevalence of 11q22 mutations in HIV positive MM patients as well as its significance were not described.

Owing to limitation of local FISH probe design, it is not possible to determine whether the additional mutations detected with del 11q22 testing are associated with the Ataxia-telangiectasia mutated (ATM) gene or other subset mutations.”

---

## [Decision Letter · Decision Letter 1]

5 Jun 2023

Profile and Outcome of Multiple Myeloma with and without HIV treated at a Tertiary Hospital in KwaZulu-Natal, South Africa

PONE-D-23-09091R1

Dear Dr. Chili,We’re pleased to inform you that your manuscript has been judged scientifically suitable for publication and will be formally accepted for publication once it meets all outstanding technical requirements.

Kind regards,

Zivanai Cuthbert Chapanduka, MBChB (M.D)

Academic Editor

PLOS ONE

Additional Editor Comments (optional):

Dear Dr Chili and Colleagues.

Thank you for submitting your research to Plos1.

Reviewers have recommended that your manuscript be published. Congratulations.

Further communication regarding the publication process, will come from the relevant Plos1 offices.

All the best.

Zi Chapanduka.

Reviewers' comments:

Reviewer's Responses to Questions

**Comments to the Author**

1. If the authors have adequately addressed your comments raised in a previous round of review and you feel that this manuscript is now acceptable for publication, you may indicate that here to bypass the “Comments to the Author” section, enter your conflict of interest statement in the “Confidential to Editor” section, and submit your "Accept" recommendation.

Reviewer #1: All comments have been addressed

Reviewer #2: All comments have been addressed

2. Is the manuscript technically sound, and do the data support the conclusions?

Reviewer #1: Yes

Reviewer #2: Yes

3. Has the statistical analysis been performed appropriately and rigorously? 

Reviewer #1: Yes

Reviewer #2: Yes

4. Have the authors made all data underlying the findings in their manuscript fully available?

Reviewer #1: Yes

Reviewer #2: Yes

5. Is the manuscript presented in an intelligible fashion and written in standard English?

Reviewer #1: Yes

Reviewer #2: Yes

6. Review Comments to the Author

Reviewer #1: (No Response)

Reviewer #2: All my queries were addressed. This is a comprehensive manuscript in an area that has not been well represented in literature. Follow up studies recommended.

7. PLOS authors have the option to publish the peer review history of their article (what does this mean?). If published, this will include your full peer review and any attached files.

Reviewer #1: **Yes: **Dr Ibtisam Abdullah

Reviewer #2: **Yes: **Leonard Mutema

---

## [Editor Report · Acceptance letter]

22 Jun 2023

PONE-D-23-09091R1 

Profile and outcome of multiple myeloma with and without HIV treated at a tertiary hospital in KwaZulu-Natal, South Africa 

Dear Dr. Chili:

I'm pleased to inform you that your manuscript has been deemed suitable for publication in PLOS ONE. Congratulations! Your manuscript is now with our production department. 

Kind regards, 

on behalf of

Professor Zivanai Cuthbert Chapanduka 

Academic Editor

PLOS ONE